# VISUAL TIMING FOR SOUND SOURCE DEPTH ESTIMATION IN THE WILD

## ABSTRACT

Depth estimation enables a wide variety of 3D applications, such as robotics and autonomous driving. Despite significant work on various depth sensors, it is challenging to develop an all-in-one method to meet multiple basic criteria. In this paper, we propose a novel audio-visual learning scheme by integrating semantic features with physical spatial cues to boost monocular depth with only one microphone. Inspired by the flash-to-bang theory, we develop FBDepth, the first passive audio-visual depth estimation framework. It is based on the difference between the time-of-flight (ToF) of the light and the sound. We formulate sound source depth estimation as an audio-visual event localization task for collision events. To approach decimeter-level depth accuracy, we design a coarse-to-fine pipeline to push the temporary localization accuracy from event-level to millisecond-level by aligning audio-visual correspondence and manipulating optical flow. FBDepth feeds the estimated visual timestamp together with the audio clip and object visual features to regress the source depth. We use a mobile phone to collect 3.6K+ video clips with 24 different objects at up to $60m$. FBDepth shows superior performance especially at a long range compared to monocular and stereo methods.

## 1 INTRODUCTION

Depth estimation is the fundamental functionality to enable 3D perception and manipulation. Although there have been significant efforts on developing depth estimation methods with various sensors, current depth estimation schemes fail to achieve a good balance on multiple basic metrics including accuracy, range, angular resolution, cost, and power consumption.

Active depth sensing methods actively emit signals, such as LiDAR (Caesar et al., 2020), structured-light (Zhang, 2012), mmWave (Barnes et al., 2020), ultrasound (Mao et al., 2016), WiFi (Vasisht et al., 2016). They compare the reflected signal with the reference signal to derive time-of-flight (ToF), phase change, or Doppler shift to estimate the depth. Active methods can achieve high accuracy because of the physical fundamental and well-designed modulated sensing signals. Lidar is the most attractive active senor due to its large sensing range and dense point cloud. However, the density is not sufficient enough to enable a small angular resolution. Therefore, the points are too sparse to be recognized at a long distance. Besides, the prohibitive cost and power consumption limit the availability of Lidar on general sensing devices.

Passive depth sensing takes signals from the environment for sensing directly. It commonly uses RGB monocular camera (Bhoi, 2019; Laga et al., 2020), stereo camera (Cheng et al., 2020), thermal camera (Lu & Lu, 2021), or multi-view cameras (Long et al., 2021a). These sensors can achieve pixel-wise angular resolution and consume pretty less energy due to omitting the signal emitting. Among them, stereo matching can effectively estimate the disparity and infer a dense depth map since it transforms the spatial depth to the visual disparity based on the solid physical law. The baseline of the stereo camera determines the effective range and accuracy. Therefore, the dimension of the stereo camera is placed as the critical trade-off with sensing metrics. Thanks to the advance in deep learning, the cheap monocular depth estimation keeps on improving performance with new network structures and high-quality datasets. However, the accuracy is still not satisfactory especially at a long range because it can only regress depth based on the implicit visual cues. It is ill-posed without any physical formulation. Besides, it heavily relies on the dataset. It requires domain adaption and camera calibration for various camera intrinsics (Li et al., 2022).

In this paper, we propose to add only one microphone to enable explicit physical depth measurement and boost the performance of a single RGB camera. It does not rely on the intrinsic of cameras and implicit visual cues. We develop a novel passive depth estimation scheme with a solid physical formulation, called Flash-to-Bang Depth (FBDepth). Flash-to-Bang is used to estimate the distance to the lightning strike according to the difference between the arrival time of a lightning flash and a thunder crack. This works because light travels a million times faster than sound. When the sound source is several miles away, the delay is large enough to be perceptible. Applying it to our context, FBDepth can estimate the depth of a collision that triggers audio-visual events. The collision event has been explored for navigation and physical search in (Gan et al., 2022), but our work is the first that uses the collision for depth estimation. Collisions are common and can arise when a ball bounces on the ground, a person takes a step, or a musician hits a drum. We identify and exploit several unique properties related to various collisions in the wild. First, the duration of a collision is short and collision events are sparse. Thus, there are few overlapped collisions. Second, though the motion of objects changes dramatically after the collision, they are almost static at the collision moment. Third, the impact sound is loud enough to propagate to a long range.

Flash-to-Bang is applied to the range of miles for human perception. Using it for general depth estimation poses several significant challenges: (i) It is inaccessible to ground truth collision time from video and audio. Video only offers up to 240 frames per second(fps), and may not capture the exact instance when the collision occurs. Audio has a high sampling rate but it is hard to detect the start of a collision solely based on the collision sound due to different sound patterns arising from collisions as well as ambient noise. (ii) We need highly accurate collision time. 1 ms error can result in a depth error of $34$ cm. (iii) Noise present in both audio and video further exacerbate the problem.

To realize our idea, we formulate the sound source depth estimation as the audio-visual localization task. Whereas existing work (Wu et al., 2019; Xia & Zhao, 2022) still focuses on 1-second-segment level localization. FBdepth performs event-level localization by aligning correspondence between the audio and the video. Apart from audio-visual semantic features as input in existing work (Tian et al., 2018; Chen et al., 2021a), we incorporate optical flow to exclude static objects with similar visual appearances. Furthermore, FBDepth applies the impulse change of optical flow to locate collision moments at the frame level. Finally, we formulate the ms-level estimation as an optimization problem of video interpolations. FBDepth succeeds to interpolate the best collision moment by maximizing the intersection between extrapolations of before-collision and after-collision flows.

With the estimated timestamp of visual collision, we regress the sound source depth with the audio clip and visual features. FBdepth avoids the requirement to know the timestamp of audio collision. Besides, different objects have subtle differences in audio-visual temporal alignment. For example, a rigid body generates the sound peak once it touches another body. But an elastic body produces little sound during the initial collision and takes several ms to produce the peak with the maximum deformation. We feed semantic features to enable the network aware of the material, size, etc.

Our main contributions are as follows:

1. To the best of our knowledge, FBDepth is the first passive audio-visual depth estimation. It brings the physical propagation property to audio-visual learning.

2. We introduce the ms-level audio-visual localization task. We propose a novel coarse-to-fine method to improve temporal resolution by leveraging the unique properties of collisions.

3. We collect 3.6K+ audio-visual samples across 24 different objects in the wild. Our extensive evaluation shows that FBDepth achieves 0.64m absolute error(AbsErr) and 2.98% AbsRel across a wide range from 2 m to 60 m. Especially, FBDepth shows more improvement in the longer range.

## 2 RELATED WORK

**Multi-modality Depth estimation.** Recent work on depth estimation has shown the benefits of fusing cameras and other active sensors. (Qiu et al., 2019; Imran et al., 2021) recover dense depth maps from sparse Lidar point clouds and a single image. (Long et al., 2021b) associates pixels with pretty sparse radar points to achieve superior accuracy. The effective range can be increased as well by Lidar-camera (Zhang et al., 2020) or Radar-camera (Zhang et al., 2021). However, these methods are still expensive in cost and power consumption.

(Gao et al., 2020; Parida et al., 2021) emit audio chirps and learn the depth map implicitly with audio reflections and a single image. However, these methods require many nearby acoustic reflectors to produce effective echos so the setup is limited in rooms. Besides, they are evaluated in an audio-visual simulator. FBDepth only uses one extra microphone to perceive natural sounds directly. It keeps the passive design of the audio but applies the physical measurement explicitly. The one-path sound propagation has a longer effective range than echoes.

**Sound source localization.** Previous systems localize sound sources with microphone arrays (Valin et al., 2003; Rascon & Meza, 2017) or one microphone with a camera (Hershey & Movellan, 1999). They intend to estimate the direction of arrival(DOA) or the distance. The DOA is inferred by the subtle difference in arrival time from the sound source to each microphone(Mao et al., 2019; Sun et al., 2022) or by semantic matching with the visual appearance if given images(Tian et al., 2018; Arandjelovic & Zisserman, 2018). The distance can be estimated by triangulation methods with multiple DOAs and room structures(Wang et al., 2021; Shen et al., 2020). Many work study the room acoustic and the distance cues from the reverberation(Singh et al., 2021; Chen et al., 2021b) but (Zahorik, 2002) shows that the reverberation has a coarse coding with the distance. Compared to these methods, FBDepth directly estimates the distance by the ToF and achieves superior accuracy to indirect triangulation methods and implicitly depth learning networks on reverberation.

**Audio-visual event localization** aims to detect and localize events in videos. (Tian et al., 2018) first propose the task and build up the audio-visual event(AVE) dataset. They apply an audio-guided visual attention mechanism to learn visual regions with the related sounding object or motions. Recent works develop dual-modality sequence-sequence framework (Lin et al., 2019) and dual attention matching mechanism (Wu et al., 2019) to leverage global features. However, the temporal event boundary is 1s-level in AVE dataset so it is split as 1s-long segments. We study the instant collision event and solve the coarse boundary problem as well.

(Gan et al., 2022) has a similar setup to ours. They use an embodied robot agent to navigate to a dropped object in 3D virtual rooms. They integrate asynchronous vision and audition and navigate to the object. The asynchronism comes from the invisibility of the object. Even though their simulator has been pretty vivid enough for semantic tasks, it has a gap in the real-world collision for the ms-level formulation. Falling objects dataset(Kotera et al., 2020), TbD dataset(Kotera et al., 2019) and TbD-3D dataset(Rozumnyi et al., 2020) explore falling motions and fast movements but they do not have audio and depth information.

**Video frame interpolation** aims to synthesize intermediate frames between existing ones of a video. Most state-of-the-art approaches explicitly or implicitly assume a simplistic linear motion. Warping-based methods (Baker et al., 2011; Park et al., 2020) apply optical flow and forward warping to shift pixels to intermediate frames linearly. Phase-based methods (Meyer et al., 2015; 2018) combine the phase information across different scales but the phase is modeled as a linear function of time. Recent methods are developed to approximate non-linear motion, such as kernel-based methods (Niklaus et al., 2017a;b), quadratic interpolation (Xu et al., 2019a), cubic motion modeling (Chi et al., 2020), etc. However, they still fail to complex non-linear motions because precise motion dynamics cannot be captured in the blind time between keyframes. Unfortunately, collisions are super non-linear and instant. Given two keyframes before and after the collision, it is ambiguous to decide whether there is a collision. Hence, these methods are not applicable. We analyze the motions before and after the collision and extrapolate optical flows to find the most potential collision position.

## 3 PROBLEM FORMULATION

We formulate the depth estimation by the physical law of wave propagation. We have:

$$\frac{d}{v} - \frac{d}{c} = T \tag{1}$$

where the depth of the sound source is $d$ and the difference between the ToF of sound and light is $T$. $c$ and $v$ denote the propagation speeds of light and sound, respectively. We can estimate $d$ based on $d = \frac{cvT}{c-v} \approx vT$ since $c \gg v$. We observe $T = T_{audio} - T_{video} + T_{hardware}$, where $T_{audio}$ and $T_{video}$ denote the event time in the audio and video recordings, respectively, and $T_{hardware}$

denotes the start time difference in the audio and video recordings. It can be small as well as have a small variance with a well-designed media system such as the Apple AVFoundation framework. We regard it as a constant unknown bias to learn.

It is impossible to label the precise $T_{video}$ and $T_{audio}$ manually. $T_{video}$ can be tagged at most frame-level. Even though many commercial cameras can support up to 240 FPS, it results in a 4-ms segment and 1.43m depth variation. Moreover, it is tough to determine the exact frame that is nearest to the collision in high FPS mode by a human being due to the constrained view of the camera. $T_{audio}$ is challenging to recognize in the wild as well. Although the audio sampling rate is high enough, we can recognize the significant early peaks instead of the first sample triggered by the collision. The best effort of segmentation is 10-ms level based on real data.

We cannot learn the timestamp with supervision. We propose a 2-stage estimation framework. The goal of the first stage is to estimate the numerical $T_{video}$. As figure 1 shows, we localize the audio-visual event in the stream and then take advantage of the unique optical flow of the collision to estimate $T_{video}$ at ms-level. In the second stage, we place the $T_{video}$ as an anchor into the audio clip and direct regress the depth with depth supervision. We make the network optimize $T_{audio}$ automatically with knowledge of the $T_{video}$, the audio waveform and visual features.

# 4 APPROACH

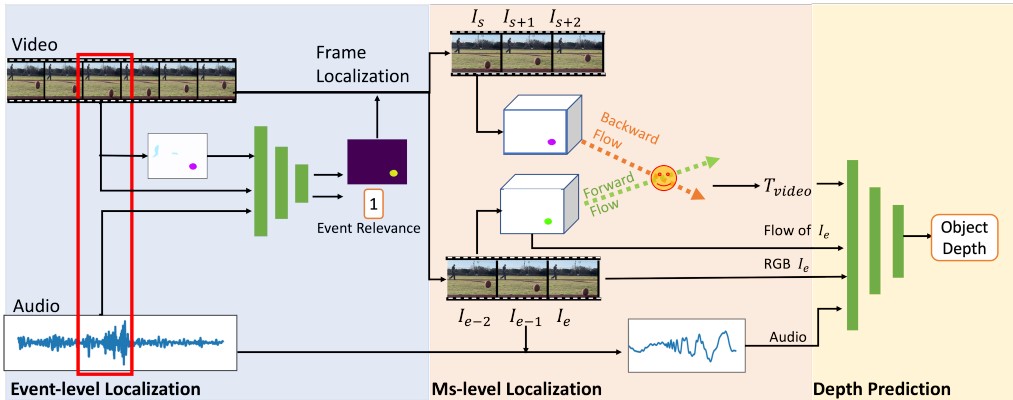

Figure 1: Model architecture. Our audio-visual depth estimation uses the video, audio and optical flow to perform the event-level localization to retrieve the collision event. It analyzes the collision flow and estimate the collision timestamp in video. It uses multiple modalities including RGB, flow, audio and the timestamp to estimate the depth.

We demonstrate a novel coarse-to-fine pipeline to localize the collision with a super temporal resolution in the video. This method does not require annotations on ms-level, which is at least two orders of magnitude finer than previous approaches. They rely on the supervision of segment annotations, such as AVE dataset with 1-second segments (Tian et al., 2018), Lip Reading Sentences 2 dataset with word-level segments (Chung & Zisserman, 2016), BOBSL with sentence-level alignments (Bull et al., 2021).

## 4.1 EVENT-LEVEL LOCALIZATION

**Audio-visual modeling for collisions.** In this step, our goal is to localize the audio-visual event for the region and the period of interest. It is similar to (Tian et al., 2018), but the unique properties of collisions bring new opportunities to learning strategy. Collisions have a significant motion than other sound sources. We can use the optical flow to inform the network of moving pixels. Besides, the impact sound is highly correlated to the rich information of objects (Gan et al., 2022), such as shape, materials, size, mass, etc. It makes audio-visual cross-matching easier than general audio-visual events so that we do not need to apply a complex scheme to learn. Another fact is that

collisions are pretty sparse temporally in the wild because the duration of collisions is extremely short. It is rare to come across overlapped collisions based on our empirical study on the basketball court. Only two frames have double collisions among all 1208 frames and a total of 203 collisions when 7 basketballs are played during a 40-s duration.

We propose a motion-guided audio-visual correspondence network (MAVNet). Similar to (Tian et al., 2018; Wu et al., 2019), MAVNet performs the cross-matching for the audio features and the RGB-F channels. Besides, it predicts audio-visual segmentation to capture whole pixels of the target object. It can achieve fine-grained audio-visual scene understanding (Zhou et al., 2022). We use the segmentation mask to filter flows of interest and perform high-resolution estimation in the next steps.

MAVNet has two backbones to deal with RGB-F channels and audio clips respectively. A U-Net (Ronneberger et al., 2015) style encoder is applied to extract the frame features conditioned by optical flows. It uses a series of convolution layers to extract visual features. Another branch is the audio encoder which takes in the time-domain signal. It has a 1D convolution layer to learn an STFT-like representation and a stack of 2D convolution layers with batch normalization to learn the semantic audio features. We replicate the audio feature, tile them to match the visual feature dimension, and concatenate the audio and visual feature maps. MAVNet has two output heads as well. the U-Net decoder applies a series of up-convolutions and skip-connections from the RGB-F encoder to fused feature maps to learn the binary segmentation mask $M$. Meanwhile, the fused feature map is fed into a binary classification head consisting of convolution layers and linear layers to predict the audio-visual event relevance $y \in \{0, 1\}$.

**Training** We use the weighted sum Binary Cross Entropy (BCE) loss as the training objective for both segmentation and the cross matching, We train all components to jointly optimize the location predictions and energy reconstruction. We minimize the total loss $\mathcal{L}_{total} = BCE(M, \hat{M}) + \lambda * BCE(y, \hat{y})$ where $\lambda$ is the hypermeter to set.

**Inference** We only use low FPS to perform MAVNet to avoid dense inference at this stage. Moreover, we do not need to activate the segmentation head until the audio clip and the frame are highly matched. Finally, MAVNet uses this audio clip to retrieve a sequence of frames including the full collision procedure.

### 4.2 FRAME-LEVEL LOCALIZATION

Given a sequence of video frames, our goal is to split them into two sets: the frames before the collision $\mathbf{V_0}$ and the frames after the collision $\mathbf{V_1}$. This essentially requires us to determine the last frame $I_e$ in $\mathbf{V_0}$ before the collision and the first frame $I_s$ in $\mathbf{V_1}$ after the collision. Thus, we locate the collision between the frame $I_e$ and $I_s$.

Based on the analysis of the physical motion, we make an important observation that can help determine $I_e$ and $I_s$. The collision results in a significant acceleration change due to the strong impulse force. Let $a_t = v_t - v_{t-1}$ and $\delta a_t = a_t - a_{t-1}$ denote the acceleration and acceleration change of frame $I_t$, respectively. $\delta a$ between $I_e$ and $I_s$ is large, while $\delta a$ between adjacent frames before or after the collision is small. If the object stops moving immediately after the collision, we take the static frame $I_{e+1}$ as $I_s$. Finally, we select the frames before $I_e$ to generate $\mathbf{V_0}$, and select the frames after $I_s$ to generate $\mathbf{V_1}$.

We use the retrieved mask in the last stage to determine the object positions in the frames and calculate the velocity, acceleration, and acceleration change. We find the $I_e$ and $I_s$ at the low FPS and then replicate the procedure for frames between $I_e$ and $I_s$ at high FPS. Finally, we locate $I_e$ and $I_s$ in the high FPS mode efficiently.

### 4.3 MS-LEVEL LOCALIZATION

To further locate the exact moment of the collision, we try to interpolate frames between $I_e$ and $I_s$ to recover the skipped frame. Unfortunately, the common assumption of frame-based interpolation is fully broken down.

**Motion consistency** is fundamental for spatio-temporal video processing. If the motion of the object is temporally stable across several frames (*e.g.*, due to a constant force), the position and pose can be predicted in the future frames as well as be interpolated between two frames. We denote it

as *motion first consistency*. However, the impact sound is caused by an impulse force, which results in a rapid change of the motion status. It breaks the motion continuity and consistency. When we observe $I_e$ and $I_s$, we cannot determine whether a collision happens or the object just flies in the air.

Luckily, the collision moment retains a new form of motion consistency. We denote it as *motion second consistency*. It reveals that the motions before and after the collision share the same intersection position. Besides, they keep the *motion first consistency* separately. Therefore, we can extrapolate the motions based on the *motion first consistency* and search for the most similar motion extrapolations by leveraging *motion second consistency*. Note that our final goal is to find the timestamp of the collision instead of the motion status at the shared position. (Kotera et al., 2019; Rozumnyi et al., 2020) try to recover the sub-frame motions and trajectories as well but they require the high FPS ground truth to guide the training. In our context, we care more about when the collision happens than what it looks like.

**Optical flow extrapolation** Optical flow is widely used for frame prediction () and interpolation (Baker et al., 2011) by warping the frame with the estimated optical flow. Because it can capture all motions of pixels and get a finer understanding of the object dynamics. The optical flow sequence is usually generated by adjacent video frames. However, it is not efficient for extrapolation. The drift of pixels in the flow requires extra iterative wrappings to align the corresponding pixels, which results in accumulation errors.

Therefore, we compute the optical flows from an anchor frame $I_a$ to the frame sequence $\mathcal{V}$ as $\{I_0, I_1, ...I_n\}$. We can estimate the flow sequence $\mathcal{F}_{a \to \mathcal{V}}$ as $\{f_{a \to 0}, f_{a \to 1}, ...f_{a \to n}\}$. As $f_{a \to n}(x, y)$ represents the movement of the pixel $I_a(x, y)$ to $I_n$, $\mathcal{F}_{a \to \mathcal{V}}(x, y)$ describes how the pixel in $I_a(x, y)$ moves across the frame sequence $\mathcal{V}$. Hence, $\mathcal{F}_{a \to \mathcal{V}}$ tracks the global motion of each pixel without iterative warpings. With the historical positions of $I_a(x, y)$ from frame $I_0$ to $I_n$, we can regress the motion of this pixel and extrapolate the flow to $f_{a \to n+\delta t}$, which is the relative pixel position to $I_{n+\delta t}$ with an arbitrary $\delta t$.

In our context, We pick $k$ consecutive frames before the collision $\mathcal{V}_{pre}$ as $\{I_{e-k+1}, I_{e-k+2}, ..., I_e\}$ and after the collision $\mathcal{V}_{post}$ as $\{I_{s+k-1}, I_{s+k-2}, ..., I_s\}$. We select the frame $I_e$ as the anchor frame. It is near the collision moment, so its motion to other frames is not dramatic and easy to be estimated. Hence, we can estimate the optical flow sequences $\mathcal{F}_{e \to \mathcal{V}_{pre}}$ and $\mathcal{F}_{e \to \mathcal{V}_{post}}$ Meanwhile, we apply the predicted segmentation mask of $I_e$ to filter the pixels of the target object. In the last step, we build up regressors $\mathcal{R}$ for each pixel's motion individually and predict future locations in any sub-frame.

**Optical flow interpolation** We have construct pixel level regressors for $\mathcal{F}_{e \to \mathcal{V}_{pre}}$ and corresponding $\mathcal{F}_{e \to \mathcal{V}_{post}}$. They can extrapolate the flow $f_{e \to e+\delta t_0}$ and $f_{a \to s+\delta t_1}$, respectively. $\delta t_0, \delta t_1$ are extrapolation steps. The optimization goal is to

$$\min_{e-s \le \delta t_1 \le 0 \le \delta t_0 \le s-e} ||f_{e \to e+\delta t_0}, f_{a \to s+\delta t_1}||_2, \text{s.t. } e + \delta t_0 < s + \delta t_1$$

The collision duration is $s + \delta t_1 - (e + \delta t_0)$, which is always more than 0. $e + \delta t_0$ is the target ms-level localization $\hat{T}_{video}$. We can apply this interpolation methodology to search the intersection of the object's center trajectory or maximize the Intersection over Union (IoU) of the object's bounding box. However, both only use several key points so they cannot achieve a fine granularity since the optical flow takes advantage of thousands of pixels.

## 4.4 Depth Regression

Based on the estimation $\hat{T}_{video}$, we directly regress the depth to fit the $T_{audio}$ and the bias $T_{Hardware}$ with the supervision of ground truth depth. We observe that the sound generation procedure varies a lot across different objects, materials, shapes, and motions. On one hand, the diverse waveforms make it impractical to measure the exact $T_{audio}$ manually. On the other hand, each specific waveform has significant implications on what is the best $T_{audio}$ corresponding to $\hat{T}_{video}$. To combat the background noise from other sources, we also feed the RGB-F crop of the target object from frame $I_e$ to the depth predictor. It includes the semantic features of the object as well as the motion status just before the collision. These cues can guide the predictor to find the waveform pattern easily.

We select a sequence of audio samples starting from $I_e$ and label some anchor samples as 1 at $\hat{T}_{video}$. It informed the audio sequence about the timestamp of the visual collision directly. We feed the enriched sequence into the 1D convolution layer to extract a 2D representation. It is followed

by two residual blocks to learn high-dimension features. Meanwhile, we use ResNet-18 (He et al., 2015) to extract the RGB-F features of the target object. We tile and concatenate the RGB-F features to the audio features along the channel dimension and append another two residual blocks to fuse the features. Finally, it is followed by a pooling layer and a fully connected layer to predict the depth. The output maps to depth by the 2D projection. We use Mean Square Error (MSE) $\mathcal{L}_{depth} = ||d, \hat{d}||_2$ as the learning objective where $d$ and $\hat{d}$ are the target depth and the predicted depth.

# 5 EXPERIMENTS

## 5.1 SETUP

**Dataset platform and collection** We use an iPhone XR with a 240-fps slow-motion mode to collect the video with audio. The audio sampling rate is 48Khz. We set a stereo camera and a Lidar together to collect ground truth. We include details of data collection in the Appendix B.

**AVD Dataset** We collect 3.6K+ raw audio-visual sequences with a single collision event as the audio-visual depth(AVD) dataset. We randomly sample raw sequences to generate train/val/eval splits, which have 2600/500/522 sequences. We augment the raw sequences by cropping one moving object from a raw video sequence and inserting it into another raw sequence with a random temporal location. Besides, we augment the raw depth with a maximum 3% random change to diversify the depth and shift audio samples accordingly to the video timestamp. More details are described in Appendix B.

**Baselines** We include three types of baseline for comparison. We compare to a monocular depth estimation method NeWCRFs (Yuan et al., 2022), a state-of-the-art(SOTA) on multiple benchmarks. We also compare to stereo matching methods including the ZED built-in ultra depth estimation SDK and a SOTA method LEAStereo (Cheng et al., 2020). We use dense depth maps collected by the Lidar to finetune the NeWCRFs and LEAStereo on images collected by the stereo camera. Despite optical flow based interpolation, we compare to interpolation using key points such as the trajectories of center or bounding boxes.

**Metrics** We use the mean absolute depth errors as $AbsErr = \frac{1}{n}\sum_{i=1}^{n}|d - \hat{d}|$, root mean square absolute relative errors $RMSE = \sqrt{\frac{1}{n}\sum_{i=1}^{n}(d - \hat{d})^2}$, $AbsRel = \frac{1}{n}\sum_{i=1}^{n}\frac{|d-\hat{d}|}{d}$ as the end-to-end performance metrics. FBDepth is a sparse depth estimation. We evaluate the depth of each target object. However, monocular and stereo baselines have dense depth estimations for all pixels of the object. We evaluate the median estimation depth with the median depth of the ground truth dense map. We provide the results over different distance ranges as close($\leq 10m$), mid($10m$-$30m$), and far($\geq 30m$). Intuitively, there is an upper bound for the temporal resolution so $AbsRel$ at close depths performs worse than at further distances.

## 5.2 RESULTS

| Method | Input | FPS | AbsErr(m) close/mid/far/all | AbsRel(%) close/mid/far/all | RMSE(m) close/mid/far/all |
|---|---|---|---|---|---|
| NeWCRFs | V | - | 0.553/1.09/3.27/1.68 | 11.1/6.74/8.64/9.48 | 0.895/1.51/5.82/3.49 |
| ZED SDK | S | - | 0.083/0.96/5.10/2.03 | 1.78/6.05/12.7/7.28 | 0.108/1.07/6.30/3.69 |
| LEAStereo | S | - | **0.067**/0.66/2.47/**0.88** | 1.48/4.24/5.98/4.09 | **0.083**/0.76/5.08/2.95 |
| FBDepth | A+V | 30 | 0.485/0.83/1.33/0.95 | 10.9/5.20/3.32/4.26 | 0.731/1.01/2.29/1.51 |
| FBDepth | A+V | 60 | 0.418/0.70/1.11/0.72 | 8.94/4.33/2.79/3.34 | 0.597/0.83/1.86/1.27 |
| FBDepth | A+V | 120 | 0.392/0.61/0.98/0.67 | 8.42/3.79/2.49/3.11 | 0.534/0.75/1.68/1.09 |
| FBDepth | A+V | 240 | 0.337/**0.58/0.95/0.64** | 7.25/**3.55/2.41/2.98** | 0.476/**0.69/1.61/1.03** |

Table 1: A comprehensive comparison for different depth estimation approaches. V, S, A represent visual, stereo, audio respectively. We input video with different frame rates as well.

Table 1 shows the results on the depth estimation. In all, FBDepth can achieve better performance on all metrics than baselines across different FPS. Several important trends can be observed. Stereo matching methods perform extraordinarily on close objects, where more clear view difference can be captured. The $AbsErr$ and $RMSE$ increase dramatically as the targets become further because

the limited baseline cannot resolve the view difference easily. In the other side, the $AbsErr$ and $RMSE$ of FBDepth grows slowly with the increasing distance while its $AbsRel$ decreases gradually. Intuitively, there is a upper bound for the temporal resolution due to the limited FPS, the lack of the accurate timestamp and the small disturbance of audio-video software. Thus FBDepth may not achieve the centimeter level easily. A Further depth can break the assumption of stereo matching methods as well as monocular methods which has a fixed depth range of training data, but FBDepth still holds the physical propagation law in this condition.

FBdepth also shows advantages on NeWCRFs. The monocular methods rely on the training set, which includes various scenarios and depths. Although we apply camera remapping with intrinsic matrix and finetuning, NeWCRFs still cannot achieve the best performance as the one in the pre-trained dataset. The implicit depth regression has difficulty in domain adaption. In the contrast, stereo methods can be directly applied to the new scenario and achieve awesome estimation because its fundamental is the explicit spatial view difference on stereo images. FBDepth applies the explicit spatial measurement and does not reply on the camera and scenarios heavily. It requires several learning models but these model can be applied to common cameras and microphones. FBDepth can be more general with a more diverse dataset. We show some visual qualitative results in the Appendix B.3. Compared to other methods, the delay between audio and video can be visually recognized, which is similar to object detection. In another word, FBDepth transforms the tough depth estimation problem to a simple interpretable problem.

## 5.3 ABLATION

In the ablation study, we show how each stage contributes to the final results.

| Method | AbsErr(m) | AbsRel(%) | RMSE(m) | Recall | Precision |
|---|---|---|---|---|---|
| event loc w/o flow | - | - | - | 87.3 | 93.7 |
| FBDepth w/o interp | 1.95 | 9.16 | 4.07 | - | - |
| FBDepth w/ center | 1.39 | 6.57 | 2.31 | - | - |
| FBDepth w/ bbox | 1.23 | 5.65 | 2.06 | - | - |
| FBDepth w/o RGB-F | 0.92 | 4.25 | 1.42 | - | - |
| FBDepth | **0.64** | **2.98** | **1.03** | 94.5 | 98.7 |

Table 2: Ablation study for FBDepth using different setups at each stage. The input is 240 FPS.

**Event-level localization** We invest how the optical flow can help detect the collision event as well as contour the object mask. We define recall and precision as the percentage of correct recognized audio-visual events in all audio-visual events and all recognized events with an IoU more than 0.5, respectively. With the flow, both recall and precision improve as the flow can work as a pre-mask to guide the network. The main failures in recall come from weak collision sounds or simultaneous collisions. The incorrect recognition is mainly due to similar objects in the frame.

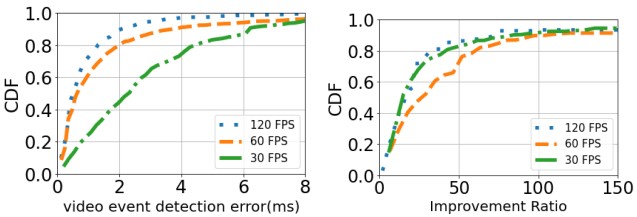

(a) Temporal error of the estimation of low FPS compared to 240 FPS

(b) Improvement ratio of temporal resolution

Figure 2: Effectiveness the video event detection in the second stage; change figure ratio

**Frame-level localization** Frame rate is most related to the frame-level stage. We observe that increasing the frame rate reduces the numerical error of FBDepth in Table 1. Especially, increasing 30 FPS to 60 FPS yields the largest improvement, and the benefit gradually tapers off with a further increase in the frame rate. We observe that 30 FPS is too slow to capture sudden movements and

fast dynamics while 60 FPS is around the borderline. It is consistent with the trend to set 60 FPS as the default video recording and playing. The motion in 120FPS and 240 FPS is even slower so it is more difficult to distinguish the frame $I_e$. The frame error is no more than the one in the low FPS mode. Thus, 120 FPS and 240 FPS bring less improvement.

**Ms-level localization** We investigate our special interpolation in two perspectives. First, we need to verify whether this method works. However, there is no ground truth timestamp so we cannot directly quantify the accuracy. We set the estimation of 240 FPS as a baseline and compare it with the estimation of lower FPS. If it can get similar numerical results from independent input, which means the algorithm is reliable. In Figure refmicro2, the median temporal error for 30, 60, and 120 FPS is 2.3ms, 0.65ms, and 0.5ms respectively. Considering the frame resolution, we can compute the improvement ratio as $\frac{frame\_duration}{temporal\_error}$. The 60 FPS has the largest 25x improvement over the frame duration. This is strong evidence that our ms-level localization is reasonable and robust.

Second, we compare the performance of depth estimation with different interpolation strategies in Table 2. We use the result from frame localization to predict the depth when there is no interpolation. The error is large since this timestamp is ambiguous for the depth prediction. Interpolation with the traces of centers or bounding boxes does not work well. A few key points cannot capture the dynamics in fine granularity.

**Depth regression** Without the RGB-F channel of the target object in depth regression, the estimation will be less robust due to the ambient sound and the background noise as shown in the Table 2

## 6  LIMITATION AND FUTURE WORKS

We classify audio-visual events into 3 categories by the quality and the quantity of visual cues during the sound production.

**Obvious visual cues during sound production** (*e.g.*collision) This is the main scenario we try to address in this paper. It requires both visible procedure and audible sound to estimate the depth. We can apply it to sports analytics, human stepping, etc. Moreover, it can collect the sparse depth point and accumulate depth points over time. According to existing work on depth completion (Long et al., 2021b; Xu et al., 2019b), adding some accurate depth points can boost the performance of monocular depth.

**Indirect visual cues during sound production** (*e.g.*speech, playing the piano) This scenario is challenging but common every day. They do not show the vibration visually. Fortunately, there are still lots of visual cues. Existing work on speech synthesis with lip motion(Ephrat & Peleg, 2017), and music generation with pose(Gan et al., 2020) indicates the strong semantic relationship between video and audio. The spatial correlation still holds here. We propose to apply a high-resolution multi-frame alignment between the video and audio to find the accurate propagation delay.

**No visual cues during sound production** (e.g. car engines, mobile phone speaker) We admit that we have no idea to estimate the depth when these sound sources are static because we cannot see them at all. Luckily, we still have a chance when these sound sources move. We propose a Doppler-like formulation to associate visual cues and audio cues.

Another urgent problem is that the microphone is pretty challenging to synchronize with other sensors. Pushing the latency to the sub-ms level can boost many applications including FBDepth.

## 7  CONCLUSION

In this paper, we develop a novel depth estimation method based on the "Flash-to-Bang". By aligning the video with the audio and detecting the events from both, we can estimate the depth in the wild without calibration or prior knowledge about the environment or target. Our extensive evaluation shows that our approach yields similar errors across varying distances. In comparison, the errors of several existing methods increase rapidly with distance. Therefore, our method is particularly attractive for large distances. As part of our future work, we are interested in further enhancing the accuracy of our method, generalizing to more contexts, and using the estimated depth to the collision to estimate the depth to other objects in the scene.

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

## A    BACKGROUND OF DEPTH SENSORS

We add more details on the performance of various depth sensors on multiple criteria in Table 3 and Table 4. We especially demo the available depth sensors and corresponding APIs on iPhone Pro 13 in Table 5 as a typical example that depth estimation is well studied at the short range.

| Sensor | Device/Method | Accuracy | Range | Angular Resolution | Power | Cost |
|---|---|---|---|---|---|---|
| LiDAR | Velodyne HDL-32E (Barnes et al., 2020) | 2cm | 100m | 1.33°(V) 0.1°-0.4°(H) | 10W | > $5000 |
| structured light | Realsense D455 | 2% | 6m | pixel-level | 3.5W | $400 |
| ToF camera | Azure Kinect (Zhang, 2012) | < 1cm | 6m | pixel-level | 5.9 W | $600 |
| mmWave | Navtech CTS350-X (Barnes et al., 2020) | 4.38cm | 163m | 1.8° | 20w | > $500 |
| inaudible sound | Rtrack (Mao et al., 2019) | 2cm | 5m | object-level | 0.5W | < $10 |
| WiFi | Chronos (Vasisht et al., 2016) | 65–98 cm | 50 m | object-level | < 10W | < $50 |

Table 3: Active depth sensors. These qualitative results may not be exact to the corresponding sensors, but they are in a similar order of magnitude.

| Sensor | Device/Method | Accuracy | Range | Angular Resolution | Power | Cost |
|--------|---------------|----------|-------|--------------------|-------|------|
| camera | NeWCRFs(Yuan et al., 2022) | NYUv2: 9.52%
KITTI: 5.20% | 10m
80m | pixel-level | <1W | <$30 |
| stereo camera | ZED 2i(StereoLab, 2021) | < 2% up to 10m;
< 7% up to 30m | 40m | pixel-level | 2W | $450 |
| camera + mic | FBDepth | overall 2.98%;
>30m: 2.41% | 60m | obj-level | <1W | <$30 |

Table 4: Passive sensors and algorithms.

| Sensor | IOS API | Accuracy | Range | Usage |
|--------|---------|----------|-------|-------|
| True depth camera | builtInTrueDepthCamera | 0.5mm | 15cm-100cm | Face ID |
| Lidar | builtInLiDARDepthCamera | 3cm | 5m | 3D scanner |
| Dual camera | builtInDualCamera | low | low | Portrait mode |

Table 5: Existing IOS APIs for developers to capture the depth(Apple, 2022). They provide accurate depth estimation in the short range. Note that although there are no qualitative results available, the accuracy and range of the stereo depth are limited due to the small baseline of the dual camera.

# B  DATASET DETAILS

We describe the details to build up the data collection pipeline for this novel task and discuss the trade-off during the data collection.

## B.1  PLATFORM AND COLLISION OBJECTS

Figure 3 shows the data collection platform. It includes three devices.

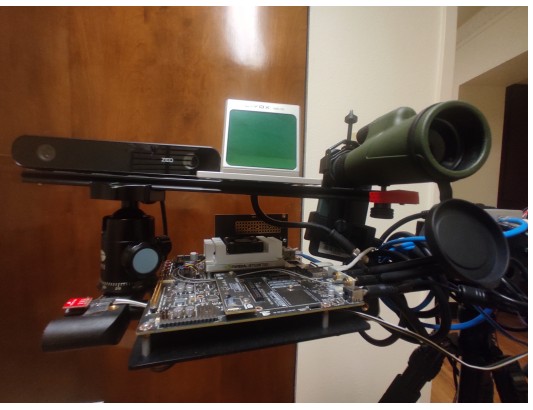

Figure 3: Data collection platform with multiple sensors

**Lidar:** We use a Livox Mid-70 Lidar(LIVOX, 2021) to collect the ground truth depth. The detection range is 90 m @ 10% reflectivity. The range precision is 2cm. Although the point rate of Mid-70 is low, it has a special non-repetitive scan pattern so that the point cloud can be very dense by accumulation. Thus, it is best to be used to collect the depth in the static scene.

**Stereo Camera:** We use a ZED 2i stereo camera(StereoLab, 2021) with a 12 cm baseline and a focal length of 4mm. The large focal length is designed to increase the maximum effective range. The image resolution is 1242 by 2208 pixels. Table 3 shows detailed performance. We use the ZED 2i camera as an important depth estimation baseline.

**Video Recorder:** A pair of a camera and a microphone can play the basic functionalities of the video recorder. However, it is very challenging to satisfy all the criteria for the audio-visual depth

estimation. In this experiment, we use an iPhone XR and record the video by the default Camera app. It has several promising advantages. First, we can record slow-motion 1080P videos with 240 fps. The frame duration is constant so that we can transform the frame number to the timestamp accurately and align it with the audio track which has a 48kHz sampling rate. Second, the audio-visual recording delay $T_{hardware}$ is small as 1 ms and has a small variance within 1 ms on the iPhone. Both specifications above are critical to the audio-visual depth but cannot be satisfied on other platforms such as Android phones. The calibration of the audio-visual recording framework is out of the scope of this work. It is unexpected that the calibration is pretty difficult based on our experience.

To capture the remote scene clearly, the telephoto lens has become indispensable in recent smart-phones. Samsung Ultra 22 can support 10x optical zoom and 100x hybrid zoom, and Pixel 6 pro has 20x zoom in all. Their zoom performance is much superior to iPhone. The iPhone XR is not equipped with a telephoto lens, so we mount an ARPBEST monocular telescope to enlarge the scene at a large distance. As shown in Figure 4, the image quality of our setup is a bit worse than the one captured by Pixel 6 Pro's telephoto lens. Thus, our setup does not provide superior image quality compared to existing commercial camera modules on smartphones. The image taken by Pixel Pro 6 is sharp but noisy while the one taken by iPhone XR with the telescope is a bit blurred. Our setup does not take advantage of the external telescope from this perspective. Overall, our setup resembles the hardware available on commercial mobile phones.

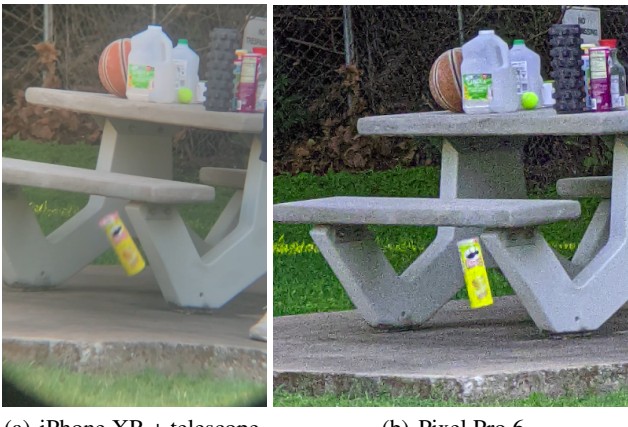

(a) iPhone XR + telescope     (b) Pixel Pro 6

Figure 4: Compare the image quality captured by our telescope setup and the commercial telephoto lens on smartphones

**Collision Objects:** In Figure 8, We use 24 objects including various masses, sizes, shapes, and six common materials: wood, metal, foam, rubber, plastic, and paper. These objects are ubiquitous every day. Besides, they do not break down during the collision.

### B.2 Collection Methodology

**Sensor setup:** We mount the Lidar, the stereo camera, and the iPhone on one slide. We perform camera Lidar calibration between the left camera of the stereo camera and the Lidar according to (Yuan et al., 2021). We use the left camera to evaluate the monocular depth estimation and use the stereo camera to evaluate the stereo depth estimation. The mobile phone changes the field of view to fit the object at different distances. Hence, its intrinsic is not constant. We use the frames recorded by iPhone only for FBDepth.

**Collision setup:** Since the point cloud is too sparse to measure the instant collision, we control the collision position to get the ground truth depth. First, we select an anchor position and measure the depth from the slide to the anchor by the Lidar. Second, we perform the collision at the anchor. For example, we throw an object to collide with the anchor or strike a hammer into the anchor or step the shoes on the anchor. Finally, the iPhone records the collision procedure. Besides, the Lidar and the stereo camera record the object placed at the anchor. They record the static object corresponding

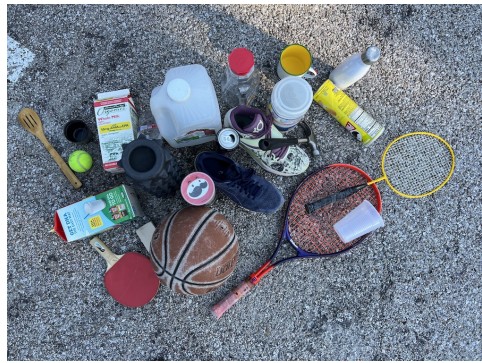

Figure 5: A set of objects are used for experiments. They cover most common items and materials in daily life

to the moving object in the video frames. We set up various anchors from 2 meters to 60 meters in different environments.

**Data Augmentation:**

After data cleaning and annotation, we get 3.6K+ raw audio-visual sequences, including 280K+ frames as te AVD dataset. Each sequence has about 40 to 120 frames and a corresponding audio clip corresponding. We use the stereo camera to capture static images and use the lidar to capture static depth maps.

We augment the raw audio-visual sequences to have more than a single collision by cropping one moving object from a raw video sequence and augmenting it to another raw sequence with a random temporal location. Meanwhile, we add up the audio sequence with the same time shift as the video. We have 10K audio-visual sequences. For the event-level localization stage, we segment an audio clip of 66.7ms including the impact sound and sample 20 frames including visible objects from each sequence and pair them as positive pairs. Negative samples pair the frame with the audio clip without impact sounds or with irrelevant impact sounds. Finally, we generate around 400K audio-visual pairs. Besides, we augment the raw depth with a maximum 3% random change to diversify the depth and shift audio samples accordingly to the video timestamp. It can solve the problem of discrete anchor depths. The change cannot be significant because the impulse response of sound is also related to depth. It requires more transformation than just shifting audio samples. We also augment images with low light, flip and rotation, and audio with diverse background noise from WHAM!Wichern et al. (2019).

### B.3 Samples and Visual Qualitative Results

We provide some samples and visual qualitative results. Considering the objects are small in the normal camera, we only show the region of interest in the RGB image and depth map. The most intuitive observation is that our approach simplifies the difficult depth estimation problem to be easily estimated from the visual samples. Humans can give a coarse estimation with the given timestamps, frames, and waveforms. However, we can have no idea to know the depth from the RGB image or stereo image visually.

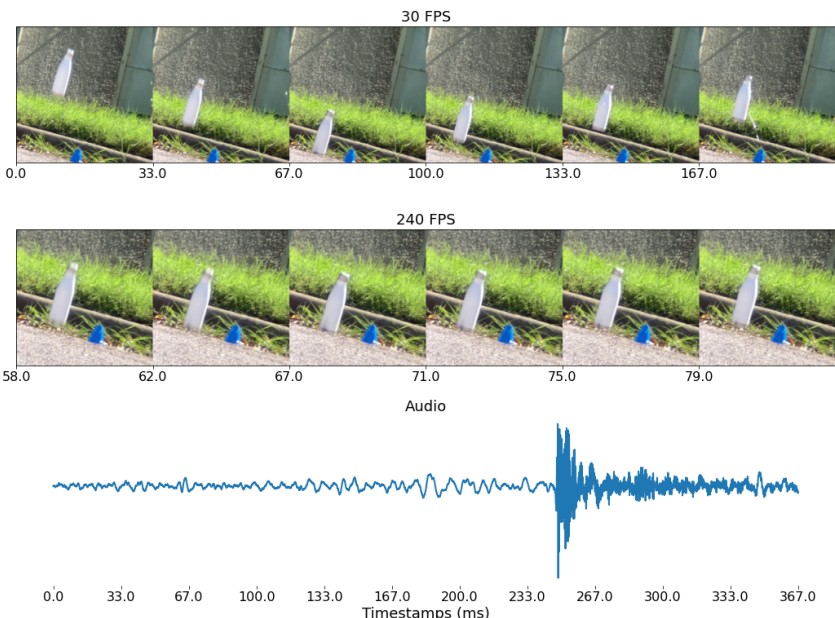

(a) Slow motion and the corresponding waveform

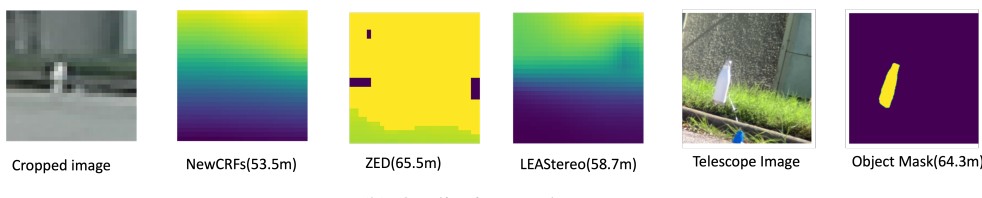

(b) Qualitative results

Figure 6: collision at 63.2m

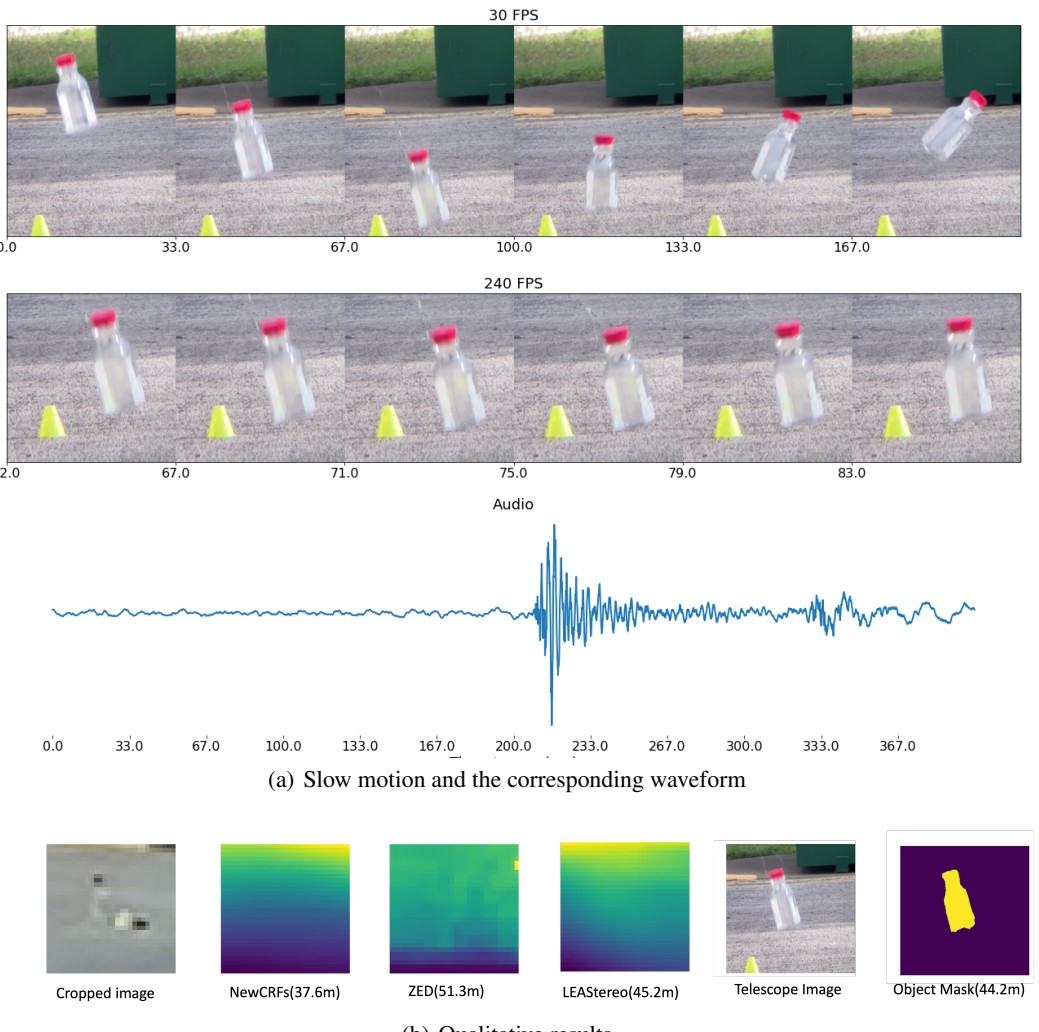

(a) Slow motion and the corresponding waveform

(b) Qualitative results

Figure 7: collision at 43.4m

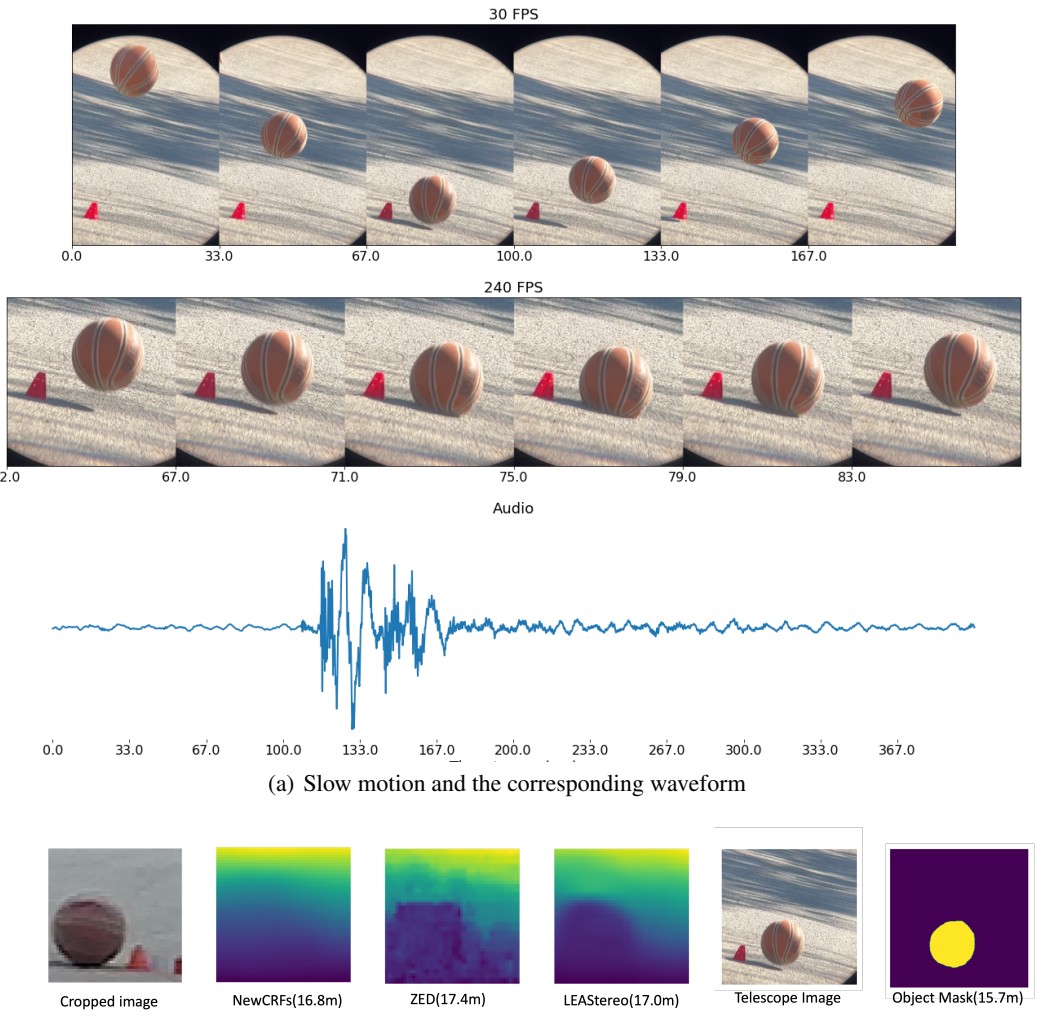

(a) Slow motion and the corresponding waveform

(b) Qualitative results

Figure 8: collision at 16.1m

