# OpenReview forum: "Visual Timing For Sound Source Depth Estimation in the Wild"
_ICLR.cc/2023/Conference — Submitted to ICLR 2023_

### Official Review · Reviewer_SYPH · 2022-10-17

**Confidence:** 3
**Correctness:** 4
**Technical Novelty And Significance:** 3
**Empirical Novelty And Significance:** 3
**Recommendation:** 6

**Clarity, Quality, Novelty And Reproducibility:**

Clarity: Generally pretty easy to follow. Some of the details of section 4 could probably be moved to supplementary materials, I think the high level idea needs to come through a bit more.

Novelty: Gan et al. have used a similar idea, but that paper was more focused on navigation. Overall this work is quite novel

Quality/Reproducibility: Both are good

I would like to see some example videos in supplementary materials. What does the video at 240fps look like, what does the audio sound like from 60m away etc.

**Strength And Weaknesses:**

Strengths
I really like this paper. Although the idea of using time of arrival differences is simple, it hasn't been explored so thoroughly before. The execution is nice. One main advantage is that this method works better with larger distances, while other depth estimation methods typically work worse with larger distances. The authors collect a clearly defined dataset of in-the-wild sounds and run experiments with that, showing promising results.

Weakness
The clear weakness is that it is not nearly as generalizable as existing methods. It almost isn't fair to compare this method to lidar, stereo etc because of how limited this method is. It only detects the depth of a single object with a clear colllision/impact. the collision must be audible but also visible. This means that many auditory events would not be applicable because they don't have a visual onset time (e.g. car engines, speech). This makes the method inapplicable for many applications of RGB->Depth. I would like to see a more clear discussion of the limitations here as well as target application scenarios the authors have in mind

**Summary Of The Paper:**

This paper presents a new approach for object depth estimation by using audio and video propagation times.

**Summary Of The Review:**

Interesting idea and well executing. Main weakness is that the method is extremely limited to very certain objects in limited scenarios. I believe it is a valid contribution, but I would like to see the limitations discussed better.

---

> ### Author Response · Authors · 2022-11-17
> **Thanks for your valuable reviews**
>
> Thanks for your valuable reviews and appreciation of our work. We are happy to share more thoughts and clarification as follow.
>
> > see a more clear discussion of the limitations here as well as target application scenarios the authors have in mind
>
> Thanks for the insight on our current approach and solution. You caught up on all the key points of the current approach and audio-visual events. We have updated it in section 6. Limitation and Future works. We have a concrete picture of the audio-visual depth estimation and some ongoing projects to solve these problems. We classify audio-visual events into 3 categories by the visual cues during the sound production.
>
> * **Obvious visual cues during sound production** (e.g. collision). This is the main scenario we try to address in this paper. It requires both visible procedure and audible sound to estimate the depth. We can apply it to sports analytics, human stepping, etc. Moreover, it can collect the sparse depth point and accumulate depth points over time. According to existing work on depth completion[1][2], adding some accurate depth points can boost the performance of monocular depth.
> * **Indirect visual cues during sound production** (e.g. speech, playing the piano). This scenario is challenging but common every day. They do not show the vibration visually. Fortunately, they still leave lots of visual cues for us to learn. Existing work on speech synthesis with lip motion[3], and music generation with pose[4] shows the strong semantic relationship between the video and the audio. The spatial correlation still holds here. We propose to apply a high-resolution multi-frame alignment between the video and audio to find the accurate propagation delay.
> * **No visual cues during sound production** (e.g. car engines, mobile phone speaker) We admit that we have no idea to estimate the depth when these sound sources are static because we cannot see them at all! Luckily, we still have a chance when these sound sources move. We also propose a doppler-like formulation to associate visual cues and audio cues.
>
> Besides, another main limitation comes from the hardware. According to my research experience on systems and my discussion with many researchers, it is always challenging to synchronize the microphone with other sensors. While multi-view cameras on different mobile phones can have submillisecond level delay[5] in a challenging setup, microphone synchronization is still at least the millisecond level. It requires a deep co-design and collaboration to resolve this problem, which can enable and boost many more applications including FBDepth.
>
> >Gan et al. have used a similar idea.
>
> Thanks for this finding. This work is really interesting. We also used the ThreeDWorld[6] simulator to simulate a collision event at the early stage. It is based on Unity engine. It applies a physical-based collision model. It can achieve 100 fps at max. However, it cannot give an accurate collision timestamp as well. It can only detect the collision at the frame level with the API of Unity. Although the simulation of collision is vivid, the colliding objects may not even touch each other during the collision moment. Finally, we directly collect the real data.
>
> > I would like to see some example videos in supplementary materials
>
> Thanks for the suggestion. I have added some examples in the Appendix B.4. I will try to add more if possible. From the samples, we can estimate the depth visually with the given timeline. Our approach simplifies the complex depth estimation problem to the same difficulty level as object detection, which can be done by human eyes directly as well.
>
> [1] Xu, Yan, et al. "Depth completion from sparse lidar data with depth-normal constraints." Proceedings of the IEEE/CVF International Conference on Computer Vision. 2019
>
> [2] Long, Yunfei, et al. "Radar-camera pixel depth association for depth completion." Proceedings of the IEEE/CVF Conference on Computer Vision and Pattern Recognition. 2021
>
> [3] Ephrat, Ariel, and Shmuel Peleg. "Vid2speech: speech reconstruction from silent video." 2017 IEEE International Conference on Acoustics, Speech and Signal Processing (ICASSP). IEEE, 2017
>
> [4] Gan, Chuang, et al. "Foley music: Learning to generate music from videos." European Conference on Computer Vision. Springer, Cham, 2020
>
> [5] Ansari, Sameer, et al. "Wireless software synchronization of multiple distributed cameras." 2019 IEEE International Conference on Computational Photography (ICCP). IEEE, 2019.
>
> [6] Gan, Chuang, et al. "The threedworld transport challenge: A visually guided task-and-motion planning benchmark towards physically realistic embodied ai." 2022 International Conference on Robotics and Automation (ICRA). IEEE, 2022.

---

> ### Author Response · Authors · 2022-12-08
> **Looking forward to your response**
>
> Dear Reviewer SYPH,
>
> Thank you again for your valuable reviews. The discussion period due(Dec 12th) is approaching. In addition to the updates in our response above, we would like to inquire if you have any more questions and suggestions on the paper and response for us to enhance the technical exposition. We welcome any feedback to improve the paper.
>
> Thank you,
>
> The authors.

---

### Official Review · Reviewer_RxUQ · 2022-10-25

**Confidence:** 3
**Correctness:** 2
**Technical Novelty And Significance:** 2
**Empirical Novelty And Significance:** 2
**Recommendation:** 3

**Clarity, Quality, Novelty And Reproducibility:**

The originality of the paper needs to be praised as it approaches depth estimation using a novel multi-modality method (audio+video).
However, the quality and the clarity of the paper still require quite some improvement to meet the bar of a top conference like ICLR.

**Strength And Weaknesses:**

### Strength
S1. This is the first work that tackles the depth estimation problem using both image and audio. The motivation for using the flash-to-bang phenomenon is quite inspiring.

S2. Besides FBDepth, the method itself, the authors also collect a new dataset that could benefit the community in this direction.

S3. The comparison with monocular and stereo-based methods shows the effectiveness of the proposed method.

### Weaknesses
W1. Although the flash-to-bang phenomenon is quite well motivated, the depth estimation, in the end, is not really making use of it directly. The network still regresses the depth directly without using Eq. (1) by audio, video, and optical flow as the inputs.

W2. The authors provide no qualitative results at all. It's pretty hard to assess how well the method works in practice without qualitative results. Both the object segmentation mask prediction and the object depth prediction are quite important to show visual results. Moreover, the authors made a very interesting new dataset (the AVD dataset), but no qualitative samples are provided at all with the submission.

W3. The method would assume a static camera if I understood correctly. It is unclear how well this method can work when the camera is moving.

W4. It is unclear how the authors make the train/val/test splits and on which split are the numbers reported.

W5. Overall the writing of the paper needs to be improved a lot. It is generally quite puzzling to understand the technical details of the paper.

**Summary Of The Paper:**

This paper presents a depth estimation method based on deep neural network, FBDepth, which makes use of both video and audio signals to estimate the object which makes collision event. FBDepth is a two-stage mehod where in the first stage it estimates the visual timing. In the second stage, FBDepth regresses the object depth based on the image, audio, visual timing, and optical flow as the inputs. To train the network, the authors collect a new dataset named  the audio-visual depth (AVD) dataset. The evaluation results show the FBDepth achieved superior results than monocular and stereo image-based depth estimation methods.

**Summary Of The Review:**

Given the current status of the paper (as I stated in the Strength And Weaknesses section), I would not recommend accepting the paper.

---

> ### Author Response · Authors · 2022-11-17
> **Thanks for your valuable review and suggestions**
>
> Thanks for your valuable reviews.  They are really important and helpful to the paper. We make the following clarifications here:
>
> >W1, regress the depth directly without using Eq. (1)
>
> Thanks for bringing up this point. Eq. (1) is the fundamental of the approach. It guides the design of the whole pipeline. We do not use the Eq. (1) directly for several reasons.
> * We cannot use the Eq. (1) directly due to the lack of ground truth timestamps and the audio-video recording latency
> * We follow the equation to formulate our problem as the audio-visual localization task and propose the coarse-to-fine pipeline to enable the final depth regression.
> * Even if a problem can be solved by equations, it is common to apply deep learning models to improve the performance, especially in building up network blocks guided by equations. For example, there are some classical signal processing algorithms to solve stereo matching and optical flow with elegant equations. These tasks are enhanced a lot with the careful design of neural networks recently. We also have a previous project on applying the deep learning models in a traditional ranging problem. The problem can be solved by a signal processing pipeline with strong mathematic fundamentals. However, the deep learning model can further improve the robustness and performance in the complex environment. Especially, we find that the model will learn a similar correlation pattern to the signal processing methods but it has a more robust representation to strengthen its performance.
>
>
> >W2, no qualitative results at all
>
> Thanks for the great suggestions. We have included the results in Appendix B.4.
>
> >W3, assume a static camera
>
> Thank you for bringing up this point. We keep the camera static during the data collection. Our contribution is  orthogonal with the camera motion. A moving camera will make the data collection more difficult.
>
> There have been a lot of existing methods to help stabilize the camera and estimate the optical flow when the camera is moving[1][2]. We can cascade them with our solution. A simple solution is to remove the camera motion estimated by the optical flow in the background from the target object optical flow.  It is also valuable to explore methods than the simple cascading.
>
> >W4, It is unclear how the authors make the train/val/test splits and on which split are the numbers reported.
>
> Thanks for the helpful suggestions. We add more details in section 5.1.  We randomly sample raw sequences to generate train/val/eval splits, which have 2600/500/522 sequences. We augument each split further.
>
> > W5, Overall the writing of the paper needs to be improved a lot . generally quite puzzling to understand the technical details of the pape
>
> Thanks for the suggestions to the writing. We have moved some parts to the Appendix, added details in the main context and polished the writing.
>
> [1] Ranjan, Anurag, et al. "Competitive collaboration: Joint unsupervised learning of depth, camera motion, optical flow and motion segmentation." Proceedings of the IEEE/CVF conference on computer vision and pattern recognition. 2019.
> [2] Almeida, Jurandy, et al. "Robust estimation of camera motion using optical flow models." International Symposium on Visual Computing. Springer, Berlin, Heidelberg, 2009.

---

> ### Author Response · Authors · 2022-12-08
> **Looking forward to your response**
>
> Dear Reviewer RxUQ,
>
> Thank you again for your valuable reviews. The discussion period due(Dec 12th) is approaching. In addition to the updates in our response above,  we would like to inquire if you have any more questions and suggestions on the paper and response for us to enhance the technical exposition. We welcome any feedback to improve the paper.
>
> Thank you,
>
> The authors.

---

### Official Review · Reviewer_Febd · 2022-10-29

**Confidence:** 3
**Correctness:** 3
**Technical Novelty And Significance:** 3
**Empirical Novelty And Significance:** 3
**Recommendation:** 6

**Clarity, Quality, Novelty And Reproducibility:**

The authors address a very interesting problem in the paper. But, the evaluation is not sufficient, and some details are missing.

The authors did not promise that they would release their dataset. Without the dataset, it is not possible to reproduce the results in the paper.

**Strength And Weaknesses:**

Pros:

+ The idea of depth estimation using audio and visual information is very interesting.  Based on the difference between the time-of-flight of the light and the sound, the authors formulate sound source depth estimation as an audio-visual collision event localization task.

+ The coarse-to-fine pipeline is technically sound. From the event level to the frame level, the authors progressively increase video event localization precision in the proposed method.

+ Compared to mono and stereo approaches, the proposed sound source depth estimation method achieves competitive performance.

Cons

- My biggest concern is about the evaluation. The current experimental results cannot fully validate the effectiveness of the proposed method. The authors provided numerical comparison results. But, visual results are totally missing in the paper.

- Some details about the collected dataset were not provided. What are object categories in the dataset? Video length? Sound source number in videos? Quality of the groundtruth depth? In addition, it would be better to provide some video samples in the dataset.

- Whether the compared monocular and stereo depth estimation methods were re-trained using the new dataset? If not, the comparison is not fair.





**Summary Of The Paper:**

Motivated by the ’flash-to-bang' phenomenon, in this paper, the authors propose a new audio-visual learning model for sound source depth estimation. In particular, the authors formulate the sound source depth estimation as an audio-visual collision event localization task.
To solve the task and increase depth estimation accuracy, a coarse-to-fine pipeline is introduced. To facilitate the research, the authors collected a new video dataset including 3600+ video clips with 24 objects. Experimental results show that the proposed approach can outperform compared mono and stereo methods.

**Summary Of The Review:**

Audio-visual sound source depth estimation is an interesting problem, and the proposed method seems technically sound to me. However, the authors did not provide sufficient experimental results to support the proposed method, and some details about the new dataset are missing. Thus, my current rating is borderline accept.

---

> ### Author Response · Authors · 2022-11-17
> **Thank you for your valuable reviews**
>
> Thank you for your valuable reviews. The comments help us to improve the paper.  We make the following clarifications here:
>
> >visual results are totally missing in this paper
>
> Thanks for the important suggestions. We have updated the paper and included visual results in Appendix B.4. The targets are small and far away. Then we crop the region of interest for observation. We can find that our solution can have a better sense of depth visually. We can estimate the audio-visual latency visually.
>
> >some details about the collected dataset
>
> Thanks for the important questions. We have updated the paper and described details of the dataset in Appendix B.1 and B.2
>
> >whether… were re-trained?
>
> Thanks for the good questions. We have clarified the baseline in section 5.1 NeWCRFs and LEAStereo has been finetuned. The ZED stereo is a pure image processing algorithm thus it does not need to be retrained.

---

> > ### Comment · Reviewer_Febd · 2022-11-23
> > **Post-rebuttal**
> >
> > Thank the authors for the response!
> >
> > After reading the updated paper and the authors' response, my concern is still there. The new visual comparison results are really confusing. The proposed method is FBDepth. But why are there only results from baselines in Figures 6, 7, and 8? I still do not how well the proposed FBDepth can predict depth by combing audio and visual inputs.
> >
> > Since I am not convinced by the rebuttal, I will keep my initial rating unchanged.

---

> > > ### Author Response · Authors · 2022-11-23
> > > **Response to the Post-rebuttal**
> > >
> > > Thanks for your valuable response. You share the concern about current visual results. We want to make more clarifications here to explain the figures.
> > > FBDepth is a sparse estimation method while baseline methods are dense estimation methods. Thus, it is not intuitive to use a depth map to represent the result of FBDepth compared to baseline methods.
> > > To show the effectiveness of FBdepth, our visual results consist of two parts:
> > > * In Figure 6.b, 7.b 8.b, we show the segmentation mask of the target object. It reflects that FBdepth finds the corresponding region of interest.
> > > * In Figure 6.a, 7.a , 8.a, we show the audio event and the visual event with the timestamps. In the 240 FPS video, we can approximate the coarse timestamps of the collision. It is between 62ms and 67ms in Figure 6.a. In the corresponding audio, we can also approximate the coarse timestamp of the collision. It is around 250ms in Figure 6.a. Thus, we can estimate the depth (0.250s - 0.064s)*343m/s = 63.8m visually.  Compared to baselines' visual results, the methodology behind FBDepth can enable humans to estimate the depth from the visualized delay.
> > >
> > > We can further label the delay in the visual results and add more description of each example in the camera-ready stage if possible. We would like to incorporate your further suggestions on the visual results. Thanks.

---

> > > ### Author Response · Authors · 2022-12-08
> > > **Looking forward to your response**
> > >
> > > Dear Reviewer Febd,
> > >
> > > Thank you again for your valuable reviews. The discussion period due(Dec 12th) is approaching. We had added more clarification for the visual results and will polish the graph in the camera-ready stage if possible. In addition to the updates in our response above, we would like to inquire if you have any more questions and suggestions on the paper and response for us to enhance the technical exposition. We welcome any feedback to improve the paper.
> > >
> > > Thank you,
> > >
> > > The authors.

---

### Official Review · Reviewer_6qDM · 2022-10-30

**Confidence:** 3
**Correctness:** 3
**Technical Novelty And Significance:** 2
**Empirical Novelty And Significance:** 3
**Recommendation:** 6

**Clarity, Quality, Novelty And Reproducibility:**

The manuscript is clearly written barring a few typos and grammatical errors here and there. The figures and tables are clear and well captioned but the language could be improved. A webpage or an appendix section with more examples and specific videos would have been a nice edition. In the current form, the manuscript leaves many questions about used videos and associated failure modes open.
The work is original and approaches an important broad idea of audio-visual cue integration for inference in the physical world through the use case of depth estimation. It seems to be reproducable if codebases and example videos are shared but those are not present in the current form.


**Strength And Weaknesses:**

Strengths -
The integration of information from sound to add visual information to yield more reliable depth is a novel and much needed idea
The use of everyday cameras and mics in smartphones, makes the results quite general and reproducible
The ablation study allows for a good understanding of incremental benefits of some of the major components of the framework

Weaknesses -
There hasn’t been any investigation of -
 - the effect of reverberation in different environments which can smear and sometimes even give different type of distance cues based of the ratio of direct-to-reverberant ratio
 - the effect of object size and material which seems to be significant considering the duration of impact can vary quite a lot based on mass and stiffness as shown by Traer et al., 2019
Many distance cues in the sounds seem to be ignored in the current framework
Lack of ground truth or a more reliable source of timestamps for verifying ms-level localization seems to be limiting
The method is only particularly attractive for longer distances


**Summary Of The Paper:**

This paper presents a novel framework, called Flash-to-Bang Depth (FBDepth), for passive sound-source depth estimation . The authors use audio-visual correspondence and optical flow manipulation to get decimeter-level depth accuracy. The proposed audio-visual depth estimation system uses video, audio and optical flow to perform event-level localization to retrieve the collision event. The main idea is based on a well-known method to estimate the distance of a lightning strike. The presented comparisons with previous depth estimation approaches show that FBDepth shows better performance than previous methods that use purely video. Although, stereo matching methods seem to be significantly better in the close range.


**Summary Of The Review:**


The proposed framework is general and performs better than most of the previous methods. It shows how using audio to augment visual information can be beneficial through the specific example task of depth estimation. That said, the use of audio information is limited to timing information usable only at large distances at the used framerates and needs more in-depth investigation. Overall, the presented work is original and tries to push the baseline on an important estimation task but it has limitations in its current form. The manuscript can greatly benefit from an investigation of distance cues present in auditory information from collisions.

---

> ### Author Response · Authors · 2022-11-17
> **Thank you for your helpful reviews and inputs(Part I)**
>
> Thank you for your review and helpful comments. They can certainly help to improve the paper and make it more accessible. We will clarify some questions below
> > Distance cues of the reverberation in the environment
>
> Thanks for your advice. We agree that reverberation can bring a lot of benefits to distance estimation. We have had experience in reverberation in previous projects. We do not explicitly consider the DRR and other reverberation features in our context for several reasons.
> * One previous work explores the localization and the reverberation in another way. We estimate the direction of arrivals of the direct path and reverberation paths and apply a delicate triangulation method to localize the speaker location. But it requires a microphone array.
> * In another work, We study reverberation and speaker distance.  It inputs reverberation speech and the image with the speaker in the environment to learn the distance and clean speech together. We also feed features such as DRR, RT60.   However, we find the reverberation features do not improve the distance estimation compared to the fusion of the visual and direct path audio directly.  In the work [1], the distance resolution is also very low.
> * Reverberation features are typically acoustic spatial features. It has a very complex formulation involving speaker location, room structure, room materials, etc. Another example is [2], whose active acoustic sensing leverages epochs to improve the depth estimation. However, the mapping between the epochs and pixels is tough and makes the estimation challenging. Thus, these features may only achieve marginal improvement compared to the current simple formulation.
> * [3] finds that the DRR provides a coarse coding of the sound source distance.
>
> > Distance cues of auditory based on object size and materials
>
> Thanks for the great suggestions. We involve these RGB and optical flow of the target objects in the depth regression. The RGB features have information on materials, object size, and object shape. The optical flow encodes the motion. They all be fed to the network to provide these distance cues.
>
> > lack of ground truth or a more reliable source of timestamps
>
> Thank you for bringing this point up. This is a pain point of current audio-video media frameworks as well as audio-video simulation.
> * As we highlight in the section Appendix B.1, the video recording of commercial devices cannot capture the collision moment. We use an indirect method to compare the estimation results of low frame rate and high frame to show the effectiveness.
> * We have tried ThreeDWorld[4] to simulate a collision event at the early stage. It is based on Unity engine. It applies a physical-based collision model. It can achieve 100 fps at max. However, it cannot give an accurate collision timestamp as well. It can only detect the collision at the frame level with the API of Unity.  Although the simulation of collision is vivid, the colliding objects may not even touch each other during the collision moment.
> * Potential solutions: It is really a challenging problem to find the ground truth timestamps in the real recording or the simulation. Two potential complex systems may be useful to get the ground truth. However, we find them still laborious, costly, and difficult to synchronize with audio recording at the current stage.
> * In future work, we plan to find a better solution to enable the ground truth timestamps.
>
> [1] Georganti, Eleftheria, et al. "Sound source distance estimation in rooms based on statistical properties of binaural signals." IEEE transactions on audio, speech, and language processing 21.8 (2013): 1727-1741.
>
> [2] Gao, Ruohan, et al. "Visualechoes: Spatial image representation learning through echolocation." European Conference on Computer Vision. Springer, Cham, 2020.
>
> [3] Zahorik, Pavel. "Direct-to-reverberant energy ratio sensitivity." The Journal of the Acoustical Society of America 112.5 (2002): 2110-2117.
>
> [4] Gan, Chuang, et al. "The threedworld transport challenge: A visually guided task-and-motion planning benchmark towards physically realistic embodied ai." 2022 International Conference on Robotics and Automation (ICRA). IEEE, 2022.

---

> > ### Author Response · Authors · 2022-11-17
> > **Thank you for your helpful reviews and inputs (Part II)**
> >
> > > only particularly attractive in the longer distance
> >
> > Thanks for bringing this point up. On one side, it is the limitation of the current solution. On the other side, it is the uniqueness of the current solution. What’s more, this point is more affected by the audio-video recording delay instead of our solution.
> > * The depth estimation is well studied at the close range. Lots of existing sensors and methods on commercial devices can perform extraordinarily for a short distance. We show 3 tables in Appendix A. We emphasize the depth sensors on the iPhone Pro series including the truth camera, Lidar scanner, and dual camera.
> > * However, the depth estimation at a longer distance still requires lots of attention. Passive methods (e.g.. monocular camera, stereo camera) have large errors. Active methods (e.g. lidar, radar) have a large angular resolution so that the small object may not have a valid measurement point at all. Multi-modality solutions (e.g. monocular + lidar) can relieve these problems but they are costly and require the alignment between the camera and the lidar.
> > * Our solution is cheap. It requires only one microphone for current camera modules. The formulation and approach are general for all distances. Unlike stereo depth, our formulation does not limit the effective range.
> > * The problem comes from the low-level hardware. The audio-video media framework cannot record them with a constant delay. Even though IOS is the best we can find, it still has a tiny variance which is smaller than 1ms. If the variance can be smaller during the data collection stage, It can be more completive on both small distances and large distances.
> >
> > > more examples of specific videos; questions about used videos and associated failure modes
> >
> > Thank you for the suggestions. We have clarified it in the section in Appendix B.4. We will add more information if possible.

---

> > > ### Comment · Reviewer_6qDM · 2022-12-04
> > > **Thanks to authors**
> > >
> > > Hi authors,
> > >
> > > Thanks for the detailed clarifications and additions to the appendix. Although I still hold some of my initial reservations about the paper, I am increasing my score by a point as I think the work with the full appedix explores an interesting direction that is worth knowing about for the other researchers in this space.

---

> > > > ### Author Response · Authors · 2022-12-04
> > > > **Thank you for your response**
> > > >
> > > > Thank you for your valuable point. We will further explore the direction with your important suggestions and encouragement.

---

### Author Response · Authors · 2022-11-23
**Paper is revised to reflect reviews and discussion**

We thank for all the reviewers for sharing the insights. The paper draft has been updated to reflect our discussion. We want to make a summary and further clarification here.

Summary of updates:
* Related work on distance cues in Section 2 sound source localization
* Limitation and future works in Section 6
* Dataset details in Section 5.1 and Appendix B.1 and B.2
* More details on baselines in Section 5.1
* Background on various depth sensors in Appendix A
* Visual results in Appendix B.3.

We need to make the clarification on visual comparison between the sparse estimation method(e.g.  FBDepth) and dense estimation methods(e.g. baseline). It is not intuitive to use a dense depth map to represent the result of FBDepth compared to baseline methods. To show the effectiveness of FBdepth, our visual results consist of two parts:
* In Figure 6.b, 7.b 8.b, we show the segmentation mask of the target object. It reflects that FBdepth finds the corresponding region of interest.
* In Figure 6.a, 7.a , 8.a, we show the audio event and the visual event with the timestamps. In the 240 FPS video, we can approximate the coarse timestamps of the collision. It is between 62ms and 67ms in Figure 6.a. In the corresponding audio, we can also approximate the coarse timestamp of the collision. It is around 250ms in Figure 6.a. Thus, we can estimate the depth (0.250s - 0.064s)*343m/s = 63.8m visually. Compared to baselines' visual results, the methodology behind FBDepth can enable humans to estimate the depth from the visualized delay.

Meanwhile, we show the region of the target cropped from the dense depth maps of baselines. It shows that we cannot intuitively percept the depth from the RGB patch or depth patch.

---

### Decision · Program_Chairs · 2023-01-20

**Decision:**

Reject

**Justification For Why Not Higher Score:**

1. Limited application domain.
2. No qualitative results.
3.  Missing many details.

**Justification For Why Not Lower Score:**

NA

**Metareview: Summary, Strengths And Weaknesses:**

This paper was reviewed by four experts in the field and received a mixed score. The main concerns are unconvincing experiments, limited application domain, and lack of clarity. The authors did a good job of rebuttal and addressed many of the concerns. However, the reviewers (including all positive ones) still feel that more work is needed to get it to the best version. AC also agrees that this work can be much stronger with additional qualitative results and demo videos. While this paper clearly has merit, the decision is not to recommend acceptance. The authors are encouraged to consider the reviewers' comments when revising the paper for submission elsewhere.



**Summary Of Ac-Reviewer Meeting:**

All reviewers agree that this paper studies a very interesting idea for sound source depth estimation. However, none of them are willing to fight for acceptance of this paper. The major concerns include the following:

1. The framework can only be used for objects with a clear collision/impact sound. It is unsure how this framework can be general enough for real-world applications.
2. No qualitative results are presented. It is very hard for reviewers to justify how well the system is.
3.  Missing many details of dataset collection and model implementations.

The final decision is a rejection.